# Detecting Deceptive Utterances Using Deep Pre-Trained Neural Networks

**Aleksander Wawer** [1,*] and **Justyna Sarzyńska-Wawer** [2]

1    Institute of Computer Sciences, Polish Academy of Sciences, Jana Kazimierza 5, 01-248 Warszawa, Poland
2    Institute of Psychology, Polish Academy of Sciences, Stefana Jaracza 1, 00-378 Warszawa, Poland; jsarzynska@psych.pan.pl
*    Correspondence: axw@ipipan.waw.pl

**Abstract:** Lying is an integral part of everyday communication in both written and oral forms. Detecting lies is therefore essential and has many possible applications. Our study aims to investigate the performance of automated lie detection methods, namely the most recent breed of pre-trained transformer neural networks capable of processing the Polish language. We used a dataset of nearly 1500 true and false statements, half of which were transcripts and the other half written statements, originating from possibly the largest study of deception in the Polish language. Technically, the problem was posed as text classification. We found that models perform better on typed than spoken utterances. The best-performing model achieved an accuracy of 0.69, which is much higher than the human performance average of 0.56. For transcribed utterances, human performance was at 0.58 and the models reached 0.62. We also explored model interpretability based on integrated gradient to shed light on classifier decisions. Our observations highlight the role of first words and phrases in model decisions, but more work is needed to systematically explore the observed patterns.

**Keywords:** deep neural networks; deception detection; lying; natural language processing

## 1. Introduction

This paper is focused on automated deception detection from texts in the Polish language. Unlike previous work that has focused on Polish, our work is oriented toward general communication instead of narrowed forms like fake news and opinion spam. We make use of the largest database of deceptive and true statements in Polish, compiled through a carefully designed data collection process controlling for topics and other variables. Our goal is to use the latest generation of neural networks to discover the most accurate methods of deception detection.

### 1.1. Related Work

The problem of detecting deceptive utterances and texts has been raised from several different research perspectives. Section 1.1.1 contains a review of research on deceptive communication from the point of view of psychology and psycholinguistics. Such approaches focus on general properties of deceptive communication. Section 1.1.2 describes another popular related area: detecting fake opinions and opinion spam, emerging from the context of online reviews and advertising. The third area, related to politics, is fake news. It is covered in Section 1.1.3.

#### 1.1.1. Psychology of Lies

Research on lie detection has been ongoing for many years but has not produced any spectacular results. In studies where participants are divided into message senders (telling the truth or lying) and judges (who evaluate the truthfulness of senders), the judges are notoriously imprecise: a meta-analysis shows that the accuracy of their judgments is only 0.54 [1]. The poor performance of human judges in this task have made the use of computer

methods such as text analysis increasingly popular. Programs that analyze English text are able to detect false statements with a 67% rate of probability [2]. The analysis of false statements emphasizes the amount of pronouns in the first person singular (I, my, etc.) and negative affect (expressions related to negative emotions) [3]. Complexity is another factor that could differentiate true statements from false statements. The process of creating a false story consumes cognitive resources [4], leading liars to tell less complex stories. In addition, stories that are less complex may focus more on simple, concrete verbs rather than evaluations and judgments. The linguistic characteristics of false statements in English are already quite well known, but there is no such research for other languages. Until now, machine learning methods to detect lying have also been used relatively rarely.

### 1.1.2. Opinion Spam

One area where deception detection methods are applied is in identifying fake product reviews. The problem is often described as detecting spam reviews, fake reviews, bogus reviews, opinion spammers, and so on.

People commonly read online opinions and reviews prior to a potential purchase. If most reviews are positive, one is likely to buy the product, and vice versa. This situation encourages the posting of fake reviews in order to promote products and potentially harm the sale of competing ones. Solving the problem of detecting opinion spam is therefore of economic importance.

In the English language, to the best of our knowledge, the first approach to automated detection of fake reviews was described in [5]. Based on the analysis of 5.8 million reviews and 2.14 million reviewers from amazon.com, the authors confirmed that opinion spam in reviews is indeed widespread.

The number of studies involving automated deception detection from texts in the Polish language is very limited.

One such study looked at the opinion spam detection problem [6]. The authors built a corpus of two types of product reviews: genuine reviews (made by real buyers) and fake reviews by two professional reviewers. The best results were obtained using non-lexical features such as the distribution of morphological classes (including part-of-speech), dictionary-based sentiment, text length, and linguistic category model (LCM) tags from a dedicated dictionary for verb categorization by their level of abstraction [7]. Overall, this set of 83 features gave an average precision of 0.83. As the authors note, the results are surprisingly good. One possible explanation for this is the rich morphology of the Polish language, which contains more information that is useful for distinguishing between true and false reviews than part-of-speech information in English. However, the results are more likely to be artificially inflated due to the small number of false review writers and consequently, a similar style of fake reviews.

### 1.1.3. Fake News

Another research area in the context of automated recognition of text veracity is fake news detection. This topic has been especially lively since the 2016 US presidential campaign. The goal of automated methods of fake news detection, as related to our work, is to analyze textual content to predict the veracity of an input text. Broadly speaking, the approaches in this category are either knowledge-rich or knowledge-lean.

The first sub-type of methods (knowledge-rich) is based on either confirming or refuting the claims made in a text piece with the use of a knowledge base such as Wikipedia [8]. This method is becoming well-known under the name of automated fact-checking and is applied in a two-step procedure. In the first step, relevant credible articles are retrieved from the knowledge base, and in the second step, inference (refuting or supporting the input claim) is performed by a neural network.

The second sub-type (knowledge-lean) infers veracity only from the input text and does not use any external source of information. Approaches here are based on various linguistic and stylometric features aided by machine learning. For example, Pérez-Rosas

et al. [9] used features such as n-grams, LIWC psycholinguistic lexicon [10], readability, and syntax. An SVM classifier was then applied to predict veracity. Another approach [11] used a similar set of features such as n-grams, parts of speech, readability scores, and the General Inquirer lexicon features [12]. The method described by [13] used a combination of features at four levels: lexicon, syntax, semantic, and discourse. They applied classification methods such as SVM (with linear kernel), random forest (RF), and XGBoost.

## 2. Materials and Methods

### 2.1. Dataset

In our study, we used statements from a project on deception detection with 400 participants aged 18–60 (F = 226; $M_{age}$ = 30.58, SD = 9.63). Overall, 4.5 percent of the subjects completed primary school, 46.5 percent had secondary education, and 49 percent had higher education. The native language of all respondents was Polish. The subjects were recruited using social media and internet advertising portals. We excluded all volunteers with psychological education and those who had any experience/training in detecting lying. Each participant was first asked to complete a short questionnaire. Based on its results, we determined which of two polarizing topics the respondent has a clearly defined position on. Topics included various social, political, economic, and sports issues (full list in Table 1). They were then asked to generate four statements. Two of them (focused on one topic) were expressed in spoken communication and recorded, and the other two were typed in an internet chat window. One statement on a particular topic was always consistent with the participant's real position, whereas the other presented an opposing viewpoint. In the case of written statements, the respondents were asked to spend at least 5 min, and in the case of oral statements, 2 min, to generate the statement. The maximum time for speaking/writing was not limited.

**Table 1.** Number of statements by topic.

| | Written | | Transcriptions | |
|---|---|---|---|---|
| **Topic** | **N** | **%** | **N** | **%** |
| Vaccinations should/ should not be compulsory | 134 | 17.6 | 92 | 12.5 |
| Polish energy should be based mainly on coal/ renewable and non-emission sources | 81 | 10.6 | 55 | 7.4 |
| People should/should not eat meat | 76 | 10 | 78 | 10.6 |
| Smartphones and social media positively/negatively affect interpersonal relationships | 54 | 7.1 | 55 | 7.4 |
| Abortion should/ should not be legal | 91 | 11.9 | 94 | 12.7 |
| God exists/does not exist | 43 | 5.6 | 50 | 6.8 |
| Robert Lewandowski is/is not the best Polish football player | 60 | 7.9 | 53 | 7.2 |
| Jerzy Zięba's treatments are/are not effective and help people heal/can harm the sick | 30 | 3.9 | 50 | 6.8 |
| Poland should/should not accept more immigrants than today | 62 | 8.1 | 40 | 5.4 |
| GM food is/is not safe and useful, and we should/should not invest in these kinds of crops | 34 | 4.5 | 34 | 4.6 |
| The political situation in Poland is going in the right/wrong direction | 32 | 4.2 | 54 | 7.3 |
| In general, most people can/cannot be trusted | 32 | 4.2 | 50 | 6.8 |
| Ewa Chodakowska is/is not the most effective personal trainer in Poland | 30 | 3.9 | 31 | 4.2 |



We gathered 1600 statements (utterances) in total, four from each participant: two real ones (one typed and one spoken) and two false ones (one typed and one spoken). Participants were asked several standardized questions while giving statements so that each one contained the same elements, such as their stance, arguments for that position, and summary. Several statements were not registered due to technical reasons, and after rejecting statements from participants who did not understand the instructions or were not Polish, we finally obtained 757 typed utterances and 730 spoken and transcribed ones.

*2.2. Human Evaluation*

Each of the statements was assessed by three judges to measure the human ability to detect deceptive utterances. The judges had no psychological or legal education or any professional experience/training in detecting lying. The judges read the original (written statements) or transcript (oral statements) texts and were asked to decide (without any time limits) whether they were true or not. Thus, the most reliable statements could be rated at 3 points (all judges considered them true) and the least credible at 0 (no judges considered them true).

*2.3. Deception Detection as Text Classification*

Our approach to the problem of automated deception detection is by means of text classification: considering the input text as a whole, the task is to distinguish whether it is true or false. In this paper we focus on using only one state-of-the-art approach to text classification, namely, deep pre-trained neural networks of the transformer architecture [14]. The dominance of this type of network is evident in the General Language Understanding Scoreboard (GLUE (http://gluebenchmark.com, accessed on 20 May 2022), [15]), a benchmark of nine different natural language understanding tasks (in English), such as, for example, question answering, paraphrasing, semantic similarity, or sentiment analysis. In our article, we select models with support for the Polish language.

*2.4. Transformer Neural Networks*

A transformer model [14] is a neural network that learns context and word meanings by tracing relationships in sequential data. Transformer models apply a technique called self-attention to detect how distant data elements in a series affect and depend on each other. An input of the attention layer is called a query. For a query, the attention layer returns the output based on a set of key–value pairs (its memory). The self-attention layer can be expressed as follows:

$$Attention(Q, K, V) = softmax(\frac{QK^T}{\sqrt{d_k}})V \qquad (1)$$

where $\sqrt{d_k}$ is the dimension of the key vector $K$ and query vector $Q$. Query ($Q$) corresponds to the current state or time step of the network. Values ($V$) are the items that network is going to pay attention to. Keys ($K$) are used to determine how much attention to pay to its corresponding value.

BERT [16] variants are transformer models with a variable number of encoder layers and self-attention heads. Encoder layers concatenate the output from all the self-attention heads, then they pass the data through a feed-forward layer.

2.4.1. BERT Models

Transformer neural network architectures emerged recently with top results in many tasks in natural language processing, such as natural language inference, question answering, sentiment analysis, and several others. BERT models [16] are perhaps the most well-known examples of transformer neural networks.

A key factor in the success of these types of models is pre-training, which allows them to gain knowledge about the natural language. This is achieved by training on relatively straightforward tasks with a huge amount of data: BERT models are pre-trained on masked

language modeling (predicting a missing word in a sentence) and next sentence prediction (predicting if one sentence follows another). By pre-training over a large corpus, as, for example, Wikipedia and thousands of books, the BERT model comes to any new task with a substantial understanding of language. Adaptation to the new task is called fine-tuning.

In our experiments, pre-trained BERT models were fine-tuned on the deception detection dataset. The final layer of the BERT-based neural network was a sigmoid activation, as our problem of distinguishing true and false texts is binary. We evaluated the classification models in a 10-fold cross-validation. In each of the folds, the training set was further randomly sub-divided into actual training (90% of data) and validation (10%) sets. The role of the validation subset was to select models: after every training epoch, each model was evaluated on this validation subset, and the best overall model was selected to avoid possible overfitting. Models were trained for 10 epochs using the Trainer API from the Transformers [17] library. We used the following parameters: 500 warm-up steps, weight decay of 0.01, the Adam optimization algorithm with weight decay (AdamW, Transformers library default setting). For each BERT variant, we used the BertForSequenceClassification implementation: Bert Model transformer with a text classification head on top (a linear layer on top of the pooled output).

We used three pre-trained models to fine-tune on the deception detection data:

- PolBERT (Polish only) [18]. The model was trained on Polish subsets of Open Subtitles and ParaCrawl, Polish Parliamentary Corpus, and Polish Wikipedia as of February 2020. The batch size was set to 4.
- HerBERT (Polish only) [19]. The model was trained on six different corpora available for Polish: CCNet Middle and Head, The National Corpus of Polish, Open Subtitles, Polish Wikipedia, and Wolne Lektury (a collection of books). The batch size was set to 8.
- BERT-base by Google (multilingual) [16]. The model was trained on texts (BooksCorpus and Wikipedia) in 102 languages, including Polish. The batch size was set to 8.

2.4.2. Universal Sentence Encoder (USE)

We also tested a different approach to using pre-trained neural networks. In this method, the fine-tuned part of the model (adapted to the new task) is separated from the constant, fixed part that computes utterance representations. In other words, text representations, also called text embedding vectors, are computed by a universally applicable neural network that does not adapt to the new task. It is another model (and not necessarily a neural network trained using the gradient descent method) that learns to classify these representations as false or true utterances.

The model we used to obtain text representations was the Universal Sentence Encoder (USE) [20]. We used version 3 of the multilingual USE, which is based on transformer architecture. This deep neural network text encoder supports 16 languages, including Polish. The input to the network is variable-length text in any of the 16 supported languages, and the output is a 512-dimensional vector. The model is intended to be used to compute universal representations of any text, suitable for tasks such as clustering, search, or information retrieval in many languages. It serves as an accurate but general-purpose vector representation of texts.

The multitask training setup of USE was based on the method described in [21]. In this technique, representations are obtained using a dual-encoder based model trained to maximize the representational similarity between sentence pairs drawn from parallel data. The representations are enhanced using multitask training and unsupervised monolingual corpora.

We used the USE text representations in radial kernel support vector machine models. The usage of this specific pairing was motivated by many papers reporting top results for moderate and small size datasets, as, for example, [22] and multiple other SemEval competitors in several tasks.

In this scenario, we used the USE network only to compute utterance embedding vectors—we did not re-train, fine-tune, or modify it for our task. We only trained an SVM model using these vectors as input. We used the NuSVC implementation from the scikit-learn package [23]. Except for the use of the radial kernel, all hyperparameters were set to their default values (not optimized in any way). The results were obtained in a 10-fold cross-validation.

### 2.5. Evaluation of Classifiers

The comparison of classification models is a complex issue and should not be limited to computing scores averaged in $k$-fold cross-validation. Although the model with the best mean performance is expected to be the best, this is not always true. Such a simple approach can be misleading, as the difference in performance might be caused by simple chance. Statistical tests should be performed to be sure that one model provides significantly higher accuracy than others. Fortunately, there is a variety of statistical methods enabling the selection of the best performing machine learning model.

We used the Cochran's Q test [24] to test the significance of the differences between several classification models. Its purpose is to determine if there are differences on a dichotomous dependent variable (which can take only one of two possible values) between more than two related groups, testing the null hypothesis that the proportion of "successes" is the same in all groups. We applied this test to the predictions of the investigated models.

Wilcoxon signed-rank tests were applied to compare two sets of predictions directly. Specifically, it was used to compare the best of the BERT models with the USE-based solution.

## 3. Results

### 3.1. Automated Predictions of Veracity: BERT Models

In the first experiment, BERT models were fine-tuned and evaluated on the same type of data: either typed or transcribed. Table 2 presents the precision, recall, F1, and accuracy of the three BERT models for typed utterances. As described in Section 2.4.1, the results were achieved in a 10-fold cross-validation with validation subsets for model selection. Table 3 presents similar data but for spoken transcribed utterances. The bottom of each table includes the results of Cochran's Q.

As indicated by the results presented in Tables 2 and 3, there were significant differences between the tested classification models. Because the $p$-value is less than 0.05, we can reject the null hypothesis and conclude that there is a difference between the classification accuracies.

Of the three models, the best performing one was HerBERT, with accuracy reaching 0.68 in the case of typed utterances. PolBERT was the second best, with an accuracy of 0.62 for typed and 0.60 for transcribed utterances. The multilingual BERT by Google failed to recognize deception, as the results are no better than the random baseline. It is possible that the multilingual pre-training does not include enough of the Polish language. As indicated in the Section 2.4.1, PolBERT and HerBERT were pre-trained using much larger, cleaner, and more carefully selected collections of texts. It is also possible that the multilingual tokenizer and embedding vectors are not as good for processing the Polish language as the monolingual equivalents in the case of PolBERT and HerBERT.

It is interesting to note that the models performed better on typed utterances. Transcriptions of spoken utterances were more difficult to learn from. This finding is interesting because spoken language leaves less time for consideration to the speaker, so deception features associated with constrained resources [4] should positively influence the performance of the classifiers. However, this did not happen or the models were not able to utilize the information.

Figure 1 illustrates 10-fold accuracy of the HerBERT model for the typed utterances, computed over 10 epochs and measured on validation sets. The results, recorded at an

interval of 500 steps, indicate that the best performance was achieved between 400 and 700 steps. The worst models exceeded the accuracy of 0.6, whereas the best ones reached 0.8.

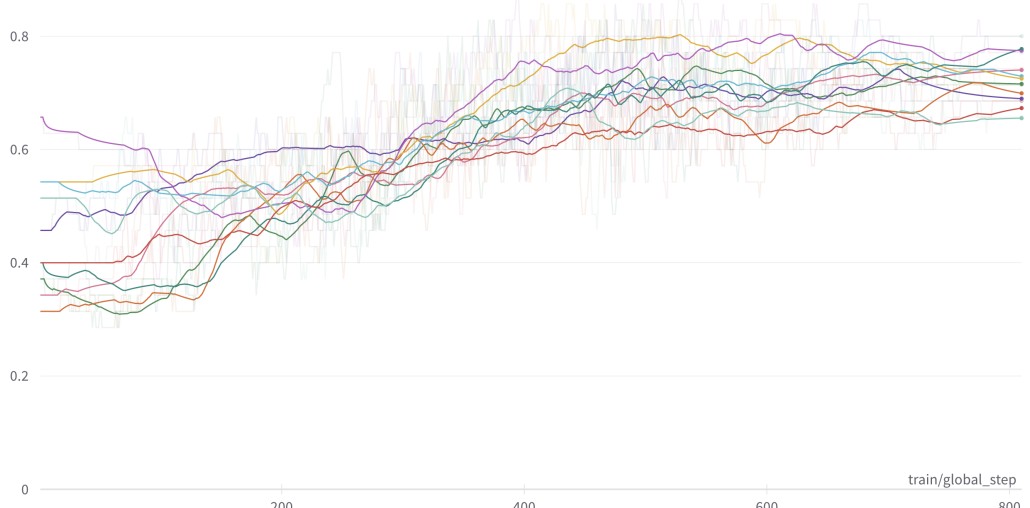

**Figure 1.** The accuracy on the validation dataset for the HerBERT model fine-tuned on typed utterances. Data for 10 folds.

We also tried to merge all texts, spoken and written, and train models on a combined dataset. On one hand, this setup could improve model training as the data size doubles. On the other, transcribed and spoken texts may be too different to extract features related to veracity. In fact, the second circumstance turned out to be true, as the models trained on the combined dataset were marginally better than the random baseline.

### 3.2. Automated Predictions of Veracity: USE+SVM

In the second experiment, the USE-encoded utterance embedding vectors were used to train SVM models with an rbf kernel. Table 4 contains the results: precision, recall, F1, and accuracy obtained in 10-fold cross-validation on typed and transcribed data.

For typed utterances, the USE and SVM combination had an accuracy of 0.69, which is only 1% better than the best BERT model (HerBERT). We performed significance tests of both models' results using Cochran Q and concluded that it is not possible to distinguish them in a significant way (typed utterances—Q: 0.067, *p*-value: 0.796; transcribed utterances—Q: 0.097, *p*-value: 0.756).

**Table 2.** Results of BERT models for typed utterances (±cross-validation standard deviations).

| Typed Utterances | | | | | |
|---|---|---|---|---|---|
| Model | Statements | Precision | Recall | F1 | Accuracy |
| PolBERT | False | $0.61 \pm 0.11$ | $0.66 \pm 0.05$ | $0.63 \pm 0.07$ | |
| | True | $0.63 \pm 0.07$ | $0.58 \pm 0.08$ | $0.61 \pm 0.05$ | |
| | | | | | $0.62 \pm 0.05$ |
| HerBERT base | False | $0.75 \pm 0.07$ | $0.53 \pm 0.09$ | $0.63 \pm 0.05$ | |
| | True | $0.64 \pm 0.12$ | $0.83 \pm 0.07$ | $0.72 \pm 0.04$ | |
| | | | | | $0.68 \pm 0.09$ |
| Google BERT base | False | $0.54 \pm 0.24$ | $0.28 \pm 0.16$ | $0.37 \pm 0.19$ | |
| | True | $0.51 \pm 0.23$ | $0.76 \pm 0.06$ | $0.61 \pm 0.06$ | |
| | | | | | $0.52 \pm 0.04$ |

Q: 52.172, *p*-value: 0.000.

**Table 3.** Results of BERT models for transcribed utterances (±cross-validation standard deviations).

| Model | Statements | Precision | Recall | F1 | Accuracy |
|---|---|---|---|---|---|
| | **Transcribed Utterances** | | | | |
| PolBERT | False | $0.65 \pm 0.13$ | $0.42 \pm 0.06$ | $0.51 \pm 0.10$ | |
| | True | $0.58 \pm 0.09$ | $0.78 \pm 0.07$ | $0.66 \pm 0.06$ | |
| | | | | | $0.60 \pm 0.06$ |
| HerBERT base | False | $0.62 \pm 0.15$ | $0.58 \pm 0.07$ | $0.60 \pm 0.05$ | |
| | True | $0.61 \pm 0.11$ | $0.65 \pm 0.09$ | $0.63 \pm 0.05$ | |
| | | | | | $0.62 \pm 0.07$ |
| Google BERT base | False | $0.53 \pm 0.33$ | $0.31 \pm 0.12$ | $0.39 \pm 0.13$ | |
| | True | $0.52 \pm 0.08$ | $0.73 \pm 0.05$ | $0.61 \pm 0.04$ | |
| | | | | | $0.52 \pm 0.06$ |

Q: 17.059, *p*-value: 0.000.

**Table 4.** USE+SVM results (±cross-validation standard deviations).

| Statements | Precision | Recall | F1 | Accuracy |
|---|---|---|---|---|
| **Typed Utterances** | | | | |
| False | $0.69 \pm 0.06$ | $0.67 \pm 0.05$ | $0.68 \pm 0.05$ | |
| True | $0.68 \pm 0.03$ | $0.70 \pm 0.05$ | $0.69 \pm 0.03$ | |
| | | | | $0.69 \pm 0.04$ |
| **Transcribed Utterances** | | | | |
| False | $0.63 \pm 0.05$ | $0.59 \pm 0.08$ | $0.61 \pm 0.05$ | |
| True | $0.62 \pm 0.05$ | $0.65 \pm 0.06$ | $0.64 \pm 0.04$ | |
| | | | | $0.62 \pm 0.03$ |

### 3.3. Human Predictions of Veracity

We checked the difficulty of distinguishing true and false statements in our dataset from the human point of view. As described before, each of the statements was assessed by three judges. We assumed a majority voting principle: a text was considered true or false when it was indicated as such by two out of three human judges. Table 5 presents the results of this evaluation. The conclusions are somewhat surprising: it turns out that, contrary to automated methods, humans perform better on transcribed utterances. However, human performance is consistently lower than transformer-based methods for both text types. The difference of 13% is more striking for typed utterances, where neural networks achieve an accuracy of 0.69 and human judges 0.56. For transcribed utterances, the difference is 4%, as neural networks reach 0.62 and human judges 0.58.

**Table 5.** Results of human judges.

| Statements | Precision | Recall | F1 | Accuracy |
|---|---|---|---|---|
| **Typed Utterances** | | | | |
| False | 0.57 | 0.53 | 0.55 | |
| True | 0.56 | 0.60 | 0.58 | |
| | | | | 0.56 |
| **Transcribed Utterances** | | | | |
| False | 0.62 | 0.41 | 0.50 | |
| True | 0.56 | 0.75 | 0.64 | |
| | | | | 0.58 |

### 3.4. Model Interpretability

In order to better understand model decisions, we applied the integrated gradient [25] method, which is an explainable AI technique. It allows one to visualize neural network input importance and explain the contribution of individual words to the model's prediction. The method needs a baseline such as a zero embedding vector for text models. It considers the straight line path from the baseline to the actual input and computes the gradients at all points along the path. Integrated gradients are obtained by accumulating these gradients. Specifically, integrated gradients are defined as the path integral of the gradients along the straight line path from the baseline to the actual input.

Table 6 contains three example texts, subject to the integrated gradient technique. Words and phrases highlighted in red illustrate word attributions toward the False class; green highlighted words and phrases lean the model toward the True class. Color saturation reflects attribution value.

**Table 6.** Examples of model interpretability. PRED: predicted class, ACT: actual class. Words and phrases that have positive correlations with the true class are shown in green, otherwise red, taking into account the degree of color saturation as an indicator of the strength of the relationship.

| | |
|---|---|
| PRED: False ACT: False | You should not eat meat because it is hard to digest and has a negative effect on our digestive system. Meat is produced in mass quantities and such production harms the environment, and killing animals is not humane. Meat can easily be replaced with ingredients of plant origin, and this would give a greater gross domestic product in our country, which is one of the largest vegetable producers in Europe. E.g., products of legume origin can replace the minced cutlets in burgers. Eating meat carries many risks as it transmits diseases and viruses that may cause a pandemic. Meat stored in our refrigerators quickly loses its value and suitability, but in stores it is soaked with various substances and salty solutions to artificially extend their saleable date. To sum up, eating meat in our economic balance is unfavorable. [..] |
| PRED: True ACT True | Poland should accept far more immigrants than before. I see no reason why the identity or purely national aspects should obscure the fundamental fact of being a human being and limit its ability to move across the surface that all people are, and at least should be, common. Since money flows in a globalized world can cross national borders, so much should people be able to do. No legal regulation can question the basic human rights to live in conditions that will be more favorable for a specific person or group of people. In discussions on migration, it is also forgotten that economic inequalities or armed conflicts between entire regions are largely the result of past colonial expansions. Therefore, it is very difficult to reliably indicate why the countries of the Occident, whose prosperity grew thanks to this type of expansion, would now close themselves to the redistribution of goods resulting from them, let alone deny the right of residence to people whose life looks like that by the previously created inequalities. |
| PRED: True ACT: False | Robert Lewandowski is a weak footballer and an amateur footballer promoted by his friend Cezary Kucharski, who built his career without technical and football skills on pairs, and his market value is overestimated thanks to skilful marketing and sponsorship activities. Summary: Robert is a weak footballer, a poor technician and a few goalscorer in Germany. Internationally little known and weak footballer among the stars not recognizable with a low contract in Bayern and Germany. |

We can observe several properties common for the model explanations:

- The most indicative for veracity are words and phrases located at the beginning of texts.

- The same phrase can attribute towards the True class in one text and towards the False class in another text. One notable example of such a phrase is "Robert Lewandowski".
- Phrases attributing to the True class are often imperative (e.g., "one/somebody should").

The above observations are preliminary and made by means of manual examination of a random sample of texts, analyzed using the integrated gradients method. Further work should be performed to verify these observations in a more systematic manner.

## 4. Conclusions

In this article, we explored the possibility of distinguishing between true and false statements by using transformer neural networks pre-trained on the Polish language and compared the results to human baselines. Our results indicate that the automated methods perform much better on this task than humans: the best of the tested approaches (USE neural network with an SVM classifier) reached an accuracy of 0.69 for typed utterances, whereas the accuracy of human judges was only 0.56. Thanks to a unique dataset, we showed that automated methods can be effective in detecting not only false statements based on facts but also human opinions, subjective feelings, and reflections.

Transcribed utterances were relatively easier to classify for humans, with an accuracy of 0.58. However, the best of the automated classification models achieved only 0.62. We can only speculate why this was the case. Most likely, the noisier spoken language contained useful veracity hints for humans and at the same time was more difficult for automated analysis. It is likely that the models were pre-trained mostly on written language, and that transcripts of oral statements were rarely found in the material used in pre-training.

The scope for further increasing the accuracy, assuming the use of text as the only data source, does not seem great. A slight improvement could possibly be achieved by the extensive optimization of hyperparameters, which is computationally expensive for transformer neural networks. A more significant improvement could be brought by multimodal representation of utterances, for example, by combining the methods used in this paper with vector representations of speech audio [26].

In the future, we plan to continue our work on model explanations to make the results more interpretable. In this context, emphasis should be placed on considering the spoken vs. transcribed nature of the utterances to shed more light on the influence of text type on veracity predictions.

**Author Contributions:** Conceptualization, A.W. and J.S.-W.; methodology, A.W. and J.S.-W.; software, A.W.; validation, A.W. and J.S.-W.; formal analysis, A.W.; investigation, J.S.-W.; resources, J.S.-W.; data curation, A.W. and J.S.-W.; writing—original draft preparation, A.W. and J.S.-W.; writing—review and editing, A.W. and J.S.-W.; visualization, A.W.; project administration, J.S.-W.; funding acquisition, J.S.-W. All authors have read and agreed to the published version of the manuscript.

**Funding:** This research was funded by The National Science Centre (Poland) grant number UMO-2017/26/D/HS6/00212 awarded by Justyna Sarzyńska-Wawer.

**Informed Consent Statement:** Informed consent was obtained from all subjects involved in the study.

**Data Availability Statement:** Datasets are available at request from the authors. Source codes for the experiments can be found in the public repository at https://github.com/alexwz/deception-detection, accessed on 20 May 2022.

**Acknowledgments:** The authors wish to thank Alksandra Pawlak and Julia Szymanowska for their support in data acquisition and Paweł Dobrowolski for text editing.

**Conflicts of Interest:** The authors declare no conflict of interest.The founders had no role in the design of the study; in the collection, analyses, or interpretation of data; in the writing of the manuscript, or in the decision to publish the results.

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
