# Peer review of "Detecting Deceptive Utterances Using Deep Pre-Trained Neural Networks"

_applsci, doi:10.3390/app12125878_

Round 1

Reviewer 1 Report

Thanks for considering my comments.

Author Response

Thank you for your valuable feedback.

Reviewer 2 Report

The authors presented the application of pretrained state-of-art deep neural network models for the detection of lies in a Polish language corpus. Although deep learning-based lie detection is of interest to readers in the natural language processing community, the authors do not propose any new classification model in this work, but rather use more commonly used deep neural network models for text classification. I do find that the authors' work is based on a new Polish language data set, however, the direct application of pre-existent neural models to classify a new language dataset can not be considered a highly novel and significant contribution to the NLP community. Therefore, I think the authors should make major revisions to their article before its publication in Applied Sciences. My comments are as follows:

1. No detailed description of the DNN models was provided in the manuscript. For example, the network structure, hyperparameter selections, training procedures, and evaluation of the generalizability of their models, etc. Adding some figures and tables to show the training procedures and performance would be beneficial for the readers of your article.

2. The authors only provided a comparison between the classification results of different DNN models and human experiments. However, I don't see any description of details for the control parameters used in the comparison experiment. 

3. The dataset contains only 1500 true and false statements, which is a relatively small corpus dataset. Moreover, the classification accuracy is a little low. Can the authors suggest any solutions to improve their classification accuracy?

Overall, this article can only be considered as an application of some pre-existent DNN models. I do not recommend its acceptance for publication in its current format. 

Author Response

Thank you for the opportunity to revise and improve the paper. We attach a file with detailed replies to each point.

Round 2

Reviewer 2 Report

Thanks for your rebuttal and it has fully resolved my questions.